# The economics of malaria control in an age of declining aid

Eric Maskin[1], Célestin Monga[2], Josselin Thuilliez [iD][3] & Jean-Claude Berthélemy[4]

This article examines financing in the fight against malaria. After briefly describing malaria control plans in Africa since 2000, it offers a stylized model of the economics of malaria and shows how health aid can help escape the malaria trap.

[1] Harvard University, Littauer Center, Room 312, 1805 Cambridge Street, Cambridge 02138 MA, USA. [2] African Development Bank, Avenue Joseph Anoma, 01 BP 1387, Abidjan 01, Côte d'Ivoire. [3] CNRS (Centre National de la Recherche Scientifique)-Centre d'economie de la Sorbonne-UMR 8174-Maison des Sciences Économiques 106-112 bd de l'Hôpital, 75642 Paris Cedex 13, France. [4] Universite Paris 1 Panth'on Sorbonne-Centre d'Economie de la Sorbonne-UMR 8174-Maison des Sciences Économiques 106-112 bd de l'Hôpital, 75642 Paris Cedex 13, France. Correspondence and requests for materials should be addressed to J.T. (email: Josselin.Thuilliez@univ-paris1.fr)

It is more than a decade since the report from the Committee on the Economics of Antimalarial Drugs, led by Nobel Laureate Kenneth Arrow was published[1]. Several studies have since shown that worldwide malaria deaths declined dramatically between 2000 and 2014[2]. This massive reduction in malaria-related mortality may have effects that reach beyond health. Improving early childhood health paves the way for greater human capital accumulation, changed fertility patterns, and faster economic development. A decline in malaria can thus generate a wide range of outcomes, many of them positive.

However, interventions across the varied epidemiological settings of Africa remain poorly understood. Two influential articles have been recently published to improve our understanding[3,4]. To assess the change in malaria prevalence from 2000 to 2015 and to quantify the effects of malaria control, these articles use a large consolidated database from field surveys and combine this database with program data to fight against malaria. They find that the prevalence of *Plasmodium falciparum* infections in Africa decreases by half between 2000 and 2015. Malaria mortality rates also fell by 57% during the period. These studies interpret these results as the effect of malaria control programs and consider that among control measures, insecticide-treated nets were the most effective. They alone contribute to 68% of the progress made. These two studies provide the best estimates to date.

However, according to the 2018 World Malaria Report, the estimated number of malaria cases worldwide was 219 million in 2017 (216 million in 2016). Data for 2015−2017 show no significant progress towards reducing the number of malaria cases worldwide and the malaria mortality reduction rate has also slowed (435,000 deaths in 2017)[5]. While these data and their quality are to be considered with caution, this stunt in the fight against malaria is attributed in part by the World Health Organization (WHO) to insufficient investments to achieve the objectives of the Global Technical Strategy for malaria 2016−2030 (GTS), namely to reduce by at least 40% malaria-specific mortality compared to 2015. Investment stagnates indeed since 2010 while the needs remain considerable and many researchers have sent out alarm signals[6,7].

This article examines financing in the fight against malaria. After briefly describing malaria control plans in Africa since 2000, it offers a stylized model of the economics of malaria and shows how foreign health aid might help escape the disease trap.

## Results

**A stylized model of malaria and health aid.** We focus on the basic reproductive number under control (PfRc), within the limits of stable *Plasmodium falciparum* transmission, which provides the potential for the disease to spread within a naive population moderated by malaria control. The estimates of PfRc were generated by the Malaria Atlas Project using a malaria transmission model to describe the relationship between PfRc and the predicted probability distribution of parasite prevalence. Figure 1 shows the average PfRc, using data from the Malaria Atlas project including 42 African countries from 2000 to 2016. Though the estimates of PfRc encapsulate uncertainty in both the underlying prevalence estimates and in the parameterization of the malaria transmission model, it is probably the best estimate at hand[3].

To understand the effects of malaria aid on the basic reproductive number under control, we start from Berthélemy and Thuilliez, who consider $R_0$ as a natural reproductive number and add individual protective decisions through a utility maximization program[8]. As usual, if $R_{0\text{Natural}}$ is lower than 1, the disease converges toward elimination, which is far from the case today (Fig. 1, green line), even when taking control programs into account. From an economic perspective, PfRc can be considered as the result of the natural basic reproductive number multiplied by the proportion of unprotected population:

$$\text{PfRc} = R_{0\,\text{Natural}}(1 - H), \qquad (1)$$

where $H$ is the aggregate protection in the population that could be thought as the use of LLINs.

The fast reduction of PfRc observed from 2000 to 2015 can be primarily considered as the positive result of Roll Back Malaria campaigns using protection tools such as LLINs—that is, an

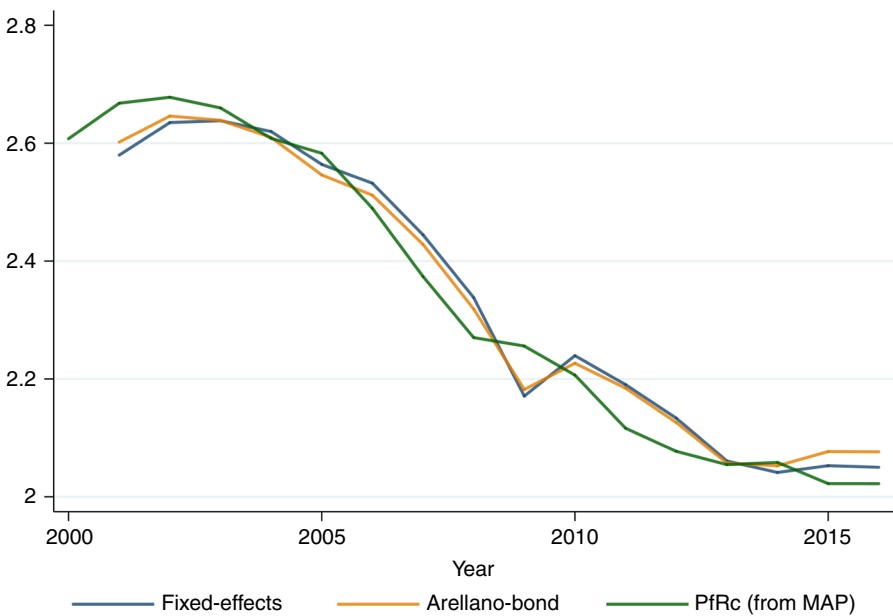

**Fig. 1** *PfRc* (geometric mean from the Malaria Atlas Project-MAP) and predicted PfRc (geometric mean from authors' predictions). The figure provides a static simulation of the model, as a simplified prediction of PfRc compared to the true values of PfRc provided by the Malaria Atlas Project. The blue line provides estimates from a standard fixed-effects model (within country). The orange line provides estimates from the Arellano−Bond model. The green line provides the Malaria Atlas Project PfRc

increase in $H$. Before the campaign, protection was relatively scarce. From an economic point of view, consider gradual increases in $H$ as the result of adopting innovation. In economic analyses, such processes of adoption follow an ordinary logistic function, which is of course S-shaped. As a result, we model the dynamics of $H$ as follows:

$$\frac{dH}{dt} = H[\alpha - \beta H], \ \alpha \leq \beta. \quad (2)$$

In the long run, $H$ converges to $\alpha/\beta$. Note that in an equality, $H$ tends to 1 and PfRc tends to 0. As a result, the rate of growth of PfRc is a nonlinear function of PfRc.

$$\frac{dPfRc}{dt} = PfRc\left[-R_{0Natural}\left(\frac{\alpha - \beta}{PfRc}\right) + \frac{\beta}{R_{0Natural}}PfRc + \alpha - 2\beta\right]. \quad (3)$$

The derivative of the growth rate of PfRc is: $\frac{R_{0Natural}(\alpha - \beta)}{PfRc^2} + \frac{\beta}{R_{0Natural}}$. It follows that for low values of $H$ (PfRc close to $R_{0Natural}$, $H$ close to 0), $dH > 0$ and $dPfRc < 0$. Symmetrically, for large values of $H$ ($R_{0Natural}$ close to 0, $H$ close to 1), $dH \leq 0$ and $dPfRc \geq 0$.

Put differently, solving Eq. (3) at time independent equilibrium yields the solution of either PfRc $= R_{0\,Natural}$ (which is unstable for $\alpha > 0$) and PfRc $= R_{0\,Natural}\left(1 - \frac{\alpha}{\beta}\right)$, which is stable for $\alpha > 0$ and $\alpha < 1$ (the condition for elimination) when $R_{0Natural}\left(1 - \frac{\alpha}{\beta}\right) < 1$. Thus, a poverty trap exists when $R_{0Natural}\left(1 - \frac{\alpha}{\beta}\right) \geq 1$.

This result is important because it shows that with the current strategy and even with a scale-up of LLINs, eliminating malaria is improbable because, for high protection coverage (large values of $H$), PfRc will tend to increase. The persistence of such a health trap, associated with $R_{0Natural}\left(1 - \frac{\alpha}{\beta}\right) > 1$—with a long run PfRc higher than 1—suggests that without intervention (through aid), it will be impossible to get out of the trap, unless very large amounts of aid are invested, assuming that aid increases adoption and coverage of LLINs—although use of interventions is less well understood—and then increases α.

The argument is not that only a big push could solve this situation, but that reducing the cost of adoption of innovations by households through aid interventions may contribute to increase LLINs' coverage—possession and use—and reduce the long-run PfRc. There are certainly other demand-side barriers and opportunity costs associated with LLINs usage that may constrain the scale-up of LLINs' coverage. Reducing malaria aid would thus be catastrophic as supported by dynamic modeling papers where a "reverse" or counterfactual scenario is explored[9].

**Illustration**. To illustrate how health aid may alleviate this issue partially, we use the panel of 42 sub-Saharan African countries to estimate the following growth model of PfRc and check whether malaria aid may enable escaping such a trap:

$$\begin{aligned}\ln(PfRc)_{Country,t} = {}& \delta + \theta_1 \ln(PfRc)_{Country,t-1} + \theta_2 \ln(PfRc)^2_{Country,t-1} \\ & + \theta_3 \text{Malaria Aid}_{Country,t} + \theta_4 \text{GDP per capita}_{Country,t} + \varepsilon_{Country,t}\end{aligned}. \quad (4)$$

Equation (4) is a discretization of the continuous equation given by Eq. (3). Malaria aid data are from WHO–World Malaria Reports and include disbursements from the Global Fund, the World Bank Booster Program, the U.S. President's Malaria Initiative, and the UK-Aid Department for International Development (DFID). GDP per capita is from WDI and PfRc is from the Malaria Atlas Project. GDP per capita proxies poverty incidence (not available on an annual basis), which can reduce the adoption of protection innovations due to poverty trap mechanisms[10,11]. We measure dPfRc/PfRc.d$t$ as a logarithmic growth rate, estimated as a quadratic function of ln(PfRc). In an extended version of Eq. (4), we add public expenditure on health from domestic sources per capita expressed in international dollars at purchasing power parity (in PPP; World Development Indicators). We next estimate the equation using a standard fixed-effects model (within country) or an Arellano–Bond model (which helps reduce the endogeneity bias inherent in dynamic fixed-effect estimations).

Results from these stylized models are provided in Table 1. Figure 1 provides a static simulation of the model, as a simplified prediction of PfRc compared to the true values of PfRc provided by the Malaria Atlas Project. Table 1 shows that the predictions of the model are confirmed. The effect of the lagged values of Ln(PfRc) are positive. Predictions also perform well and show that the level of PfRc is far above 1 in all cases. Malaria aid has a significant negative effect on PfRc in Africa, an argument in favor of sustained development assistance, but current levels of malaria aid will not be enough. Dynamic

**Table 1 Results from fixed-effects and Arellano−Bond estimates of Eq. (4)**

| Dependent var. is Ln(PfRc) | Fixed-effects model | Arellano−Bond | Fixed-effects model | Arellano−Bond |
|---|---|---|---|---|
| Lag. Ln(PfRc) | 0.817*** | 0.580*** | 0.814*** | 0.561*** |
| | (0.046) | (0.038) | (0.050) | (0.041) |
| Squared Lag. Ln(PfRc) | 0.035** | 0.065*** | 0.036** | 0.068*** |
| | (0.016) | (0.014) | (0.018) | (0.015) |
| Malaria aid per capita | −0.002* | −0.011*** | −0.003* | −0.010*** |
| | (0.002) | (0.002) | (0.002) | (0.002) |
| GDP per capita | −9.84e-06*** | −2.74e-05*** | −1.02e-05*** | −2.83e-05*** |
| | (0.000) | (0.000) | (0.000) | (0.000) |
| Domestic general government health expenditure per capita | . | . | 1.66e-05 | −6.66e-04*** |
| | | | (0.000) | (0.000) |
| Intercept | 0.122*** | 0.335*** | 0.127*** | 0.404*** |
| | (0.025) | (0.022) | (0.030) | (0.027) |
| Observations | 645 | 602 | 575 | 534 |
| Countries | 42 | 42 | 40 | 40 |
| Years | 2000−2016 | 2001−2016 | 2000−2016 | 2001−2016 |

*, ** and *** denote statistical significance at the 10, 5 and 1% level, respectively

extensions and simulations from our stylized model suggest that a level of aid of US$25–US$30 per capita would be needed to reach a PfRc of 1. This result is in accordance with other estimations made in Africa[12], where figures of up to US$20 were estimated for elimination to be possible with current interventions. It has been estimated that in the Asia-Pacific, where malaria incidence is lower, levels of around US$6–US$9 would be required for elimination[13]. Importantly, the fact that aid has a greater impact than local public health expenditures is an indication of the weakness of national systems—the expenditure financed by aid being certainly better controlled and administered and therefore more effective.

Figure 1 suggests a prevention adoption behavior that is S-shaped and has helped reduce PfRc from about 3 initially to about 2 in the long run, which would suggest a long run $H$ of about 33% on average at current levels of aid and GDP per capita. This relatively low potential adoption may be partly explained by a poverty and malaria trap mechanisms[8,14,15], in which individuals in a poverty trap would not adopt LLINs even if they are fully subsidized. In recent years the poverty incidence (at $1.90 a day) has on average been as high as 43% in sub-Saharan Africa (according to the data provided by the World Bank Povcalnet website).

## Discussion

Of course, the malaria community acknowledges that current level of international funding will not be sufficient to eliminate malaria. However, existing predictions do not illustrate the possibility and the danger of a disease trap. Moreover, the advocacy on malaria aid has not been supported by the community of economists since Sachs et al. or Arrow[1,10,11]. Our model provides a renewed analysis on malaria aid. It also uses foundations that are anchored in microeconomics to illustrate the conditions of existence of a potential malaria trap. We note that even with health aid, it is unlikely that malaria will be successfully eliminated, which might generate a negative incentive for international funders to invest in malaria control programs. This is due to several factors including the arsenal of currently available tools that may not be sufficient to control malaria to low levels of transmission, sustainability of international funding as countries develop and become wealthier, the role of public financing, malaria epidemiological specificities, among others. Moreover, combinations of tools (including LLINs, insecticides spraying, Intermittent Preventive Treatments and Artemisinin Combination Therapies) are necessary to control malaria. Our model could apply to all of these measures. However, LLINs have crystalized the debate in the economic literature and we believe our stylized model is easier to understand with LLINs alone. The unprotected population could also benefit from positive externalities through those who are protected, as demonstrated for other infectious diseases[16]. This should increase the effectiveness of both malaria aid and malaria control and reinforce our conclusions, as demonstrated elsewhere[8].

To conclude, the poverty–disease-trap mechanisms may help explain a large part of the problem. New international funding sources or innovative financing are unlikely in a near future. Poverty reduction and universal strategies would thus be a natural complement of health aid to help eliminate diseases like malaria. In light of the global economy's fragile recovery, African countries will need to reset their relationships with international aid agencies and bilateral partners and possibly look more to South–South cooperation and private–public partnerships.

Greater coordination of current aid players is therefore needed so that available resources could have higher health impact.

## Data availability

All data used for this analysis are freely accessible at Malaria Atlas Project for the PfRc (https://map.ox.ac.uk/), World Development Indicators (https://datacatalog. worldbank.org/dataset/world-development-indicators), and WHO for malaria aid (https://www.who.int/malaria/data/en/). All data were accessed on March 2018. The do file and database used for the analysis are available upon request.

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

## Acknowledgements

This study was funded by the African Development Bank Group; contract no. ECVP/SC/ 2017/012/014.

## Author contributions

E.M., C.M., J.T., and J.-C.B. designed the study and wrote the report. J.T. and J.-C.B. analyzed the data.

## Additional information

**Competing interests:** The authors declare no competing interests.

