## [Peer Review File · Nature Communications]

Reviewers' comments:

Reviewer #1 (Remarks to the Author):

Dear authors,

Thank you for submitting this paper that attempts to bring together economics and epidemiology considerations in the context of malaria. There are few studies of this kind and efforts in this field of research is appreciated.

The question raised in your paper, namely "whether it is possible to control or eliminate malaria in Africa without health aid" likely prompts the answer "no". However, even with health aid, it is unlikely that malaria will be successfully eliminated. This is due to several factors including the arsenal of currently available tools that may not be sufficient to control malaria to low levels of transmission, sustainability of international funding as countries develop and become wealthier, the role of public financing, malaria epidemiological specificities, etc.

There are several areas in your study that require clarifications:

1) rationale for considering LLIN only whilst there are mix of interventions being implemented, and evidence shows that LLIN, IRS and ACT have some control effects.

2) rationale for not considering the total envelop of funding available for malaria control, especially by ignoring governments funding in malaria control

3) lines 35-36: reference to the "efficiency" of malaria control interventions is not clear. should one read instead "effective" when thinking about the impact that malaria interventions had on malaria control in the past?

4) Line 42: flat funding is not the only factor that may contribute to stagnation in malaria control

5) line 81: not only aid increases LLIN adoption - it is likely to contribute to high level of coverage although use of interventions is less well understood.

6) eq 1: what about positive externalities of using LLIN for the "unprotected" population (those not sleeping under a net who potentially benefit from LLIN use in their community)?

7) line 83: what is the rationale that reducing the "cost of adoption of innovations to households through aid interventions may contribute to reduce transmission"? there are certainly demand side barriers, although in the case of LLIN that have largely been distributed free of charge through mass campaign, this argument seems not well articulated with the rest of the paper. Similar comment in relation to line 123 about LLIN subsidy.

8) equation 4: GDP per capita and international funding are closely linked as the main international funding source uses per capita income for funding eligibility and volume of aid provided - could the rationale for this specification be explained in more details?

9) line 106: it is acknowledged by the malaria community that level of current international funding will not be sufficient to eliminate malaria - not clear what new findings are provided here.

10) line 131: agree that coming decades will require greater domestic ownership of health systems. Could this aspect be demonstrated by analysing aspects of government funding and health systems required characteristics for improved malaria control?

11) line 132. it seems important to make the case of greater coordination of current aid players such that available resources can have higher health impact. New international funding sources or innovative financing are unlikely in near future.

Reviewer #2 (Remarks to the Author):

If a deadly disease like Plasmodium falciparum malaria is both preventable and treatable, then it is reasonable to ask the question "why do we still have malaria"? A visual comparison of a world map of malaria burden with one of poverty will illustrate a potential answer to this question. This article succinctly examines the existence of a "disease trap", essentially a public health version of a poverty trap for malaria. At a time when malaria financing is uncertain and recent increases in its burden threaten to derail a global campaign towards eradication, this article is both timely and important.

I think the authors are right to put a figure on the per capita level of aid required. These figures are subject to massive uncertainty, but a ballpark figure is useful for global financing discussions can be refined using microeconomic approaches at national and regional levels.

I strongly recommend the publication of this article.

A few very minor points of clarification follow:

1. Preface: "whether possible to control" should be "whether it is possible to control"
2. Equations 2 & 3. The stylized model is transparent and easy to follow. I am more familiar with seeing differential equations as $dX/dt = [a \text{ function of } X \text{ and parameters}]$. Is there a reason why equations (2) and (3) are not expressed in this form?
3. Line 75. I do not understand how $R_{0Natural}$ could be a function of H except in the mathematical sense of making it the subject of the formula for $PfRc$ given in equation (1). The basic reproduction number of any disease is defined as the expected number of new infections derived from a single infection in a fully susceptible population. For malaria it is therefore a measure of the receptivity of a setting and would not vary with changes in H . Only $PfRc$ would vary with changes in H whereas $R_{0Natural}$ would be fixed. The authors should explain this statement more clearly or remove it.
4. Line 79. Solving Equation 3 at time independent equilibrium yields the solution of either $PfRc = R_{0Natural}$ (which is unstable for $\alpha > 0$) and $PfRc = R_{0Natural} * (1 - (\alpha/\beta))$, which is stable for $\alpha > 0$ and less than unity (the condition for elimination) when $R_{0Natural} * (1 - (\alpha/\beta)) < 1$. Thus, the poverty trap is for $R_{0Natural} * (1 - (\alpha/\beta)) \geq 1$. So, unless I have missed something, $PfRc$ will converge to $R_{0Natural} * (1 - (\alpha/\beta))$ not α/β as stated on line 79.
5. Line 85. "reducing health aid would be catastrophic". This statement is supported by dynamic modelling papers where a "reverse" or counterfactual scenario is explored (EG: Griffin et al. Potential for reduction of burden and local elimination of malaria by reducing Plasmodium falciparum malaria transmission: a mathematical modelling study. The Lancet Infectious Diseases, 2016). The authors may wish to refer to some of these in support of this statement in the discussion section.
6. Equation 4. This is a discretization of the continuous system given by equation (3). It would help to state this and to relate θ_i , δ and ϵ to the parameters of equation (3), α , etc.
7. Line 106-7. "Dynamic predictions suggest that a level of aid of US\$ 25 to \$30 per capita would be needed to reach a $PfRc$ of 1." What dynamic predictions? Does this come from the analysis presented or from other analyses in the literature? Please clarify.
8. Line 106-7. "Dynamic predictions suggest that a level of aid of US\$ 25 to \$30 per capita would be needed to reach a $PfRc$ of 1." This seems about right for highly endemic African settings (Walker et al. Estimating the most efficient allocation of interventions to achieve reductions in

Plasmodium falciparum malaria burden and transmission in Africa: a modelling study. The Lancet Global Health, 2016) where figures of up to US\$ 20 were estimated where elimination would be possible with current interventions, leaving some settings, predicted as requiring innovation to achieve elimination, without a predicted cost. It has been estimated that in the Asia-Pacific, where malaria incidence is lower, indicate levels of around US\$ 6 to US\$ 9 would be required for elimination

(<http://www.shrinkingthemalariamap.org/sites/www.shrinkingthemalariamap.org/files/content/page/an-investment-case-for-eliminating-malaria-in-asia-pacific.pdf>).

9. Table 1: A potential next step is to visualize the per capita investment requirements derived from the model and the MAP estimates of PfRc on a world map. Have the authors considered this? Beyond the scope of this article perhaps.

Lisa Jane White

Reviewer #1 (Remarks to the Author):

Thank you for your comments, which we found very useful in the revision process. Below please find our responses (your comments are in blue and italics).

Dear authors,

Thank you for submitting this paper that attempts to bring together economics and epidemiology considerations in the context of malaria. There are few studies of this kind and efforts in this field of research is appreciated.

The question raised in your paper, namely "whether it is possible to control or eliminate malaria in Africa without health aid" likely prompts the answer "no". However, even with health aid, it is unlikely that malaria will be successfully eliminated. This is due to several factors including the arsenal of currently available tools that may not be sufficient to control malaria to low levels of transmission, sustainability of international funding as countries develop and become wealthier, the role of public financing, malaria epidemiological specificities, etc.

Many thanks for this clarification. We fully agree with this statement and we added a comment in the discussion. For modelling purpose, and for the sake of clarity, we focused on this particular aspect but this comment is now underlined in the new version of the manuscript.

There are several areas in your study that require clarifications:

1) rationale for considering LLIN only whilst there are mix of interventions being implemented, and evidence shows that LLIN, IRS and ACT have some control effects.

LLINs have been shown to be one of the most effective tools (Bhatt, S. et al. 2015, *Nature*). It is true however, that it is the combinations of tools (including LLINs, IRS, IPT and ACTs) that help fighting malaria. Our model could apply to all of these measures, including intermittent preventive treatment (IPT). However, LLINs have crystalized the debate in the economic literature and we believe it is easier to understand our point with this simplified example. We added a discussion clarifying that the model holds if we include other control measures.

2) rationale for not considering the total envelop of funding available for malaria control, especially by ignoring governments funding in malaria control

General health governments funding have now been included in our framework and this does not affect the conclusions of our analysis (they include malaria funding). The reason for this is probably that the Abuja declaration was only partially followed by governments and that GDP per capita aimed to control for total domestic resources. Please note that the variable GDP per capita is highly correlated with domestic general government health expenditure per capita in our data. The table with domestic general government expenditure per capita is provided below and we also made this change in the manuscript. There are two issues with this new table. First, the Kingdom of Swaziland is now the Kingdom of Eswatini and the World bank has not updated all the data for this particular country in the WDI. As a result, we lose one country. Second, this particular variable contains a number of missing values and we lose some observations in the process and another country.

Dependent Var. is Ln(PfRc)	Fixed-Effects Model	Arellano- Bond	Fixed-Effects Model	Arellano- Bond
Lag. Ln(PfRc)	0.817*** (0.046)	0.580*** (0.038)	0.814*** (0.050)	0.561*** (0.041)
Squared Lag. Ln(PfRc)	0.035** (0.016)	0.065*** (0.014)	0.036** (0.018)	0.068*** (0.015)
Malaria Aid per capita	-0.002* (0.002)	-0.011*** (0.002)	-0.003* (0.002)	-0.010*** (0.002)
GDP per capita	-9.84e-06*** (0.000)	-2.74e-05*** (0.000)	-1.02e-05*** (0.000)	-2.83e-05*** (0.000)
Domestic general government health expenditure per capita	.	.	1.66e-05 (0.000)	-6.66e-04*** (0.000)
Intercept	0.122*** (0.025)	0.335*** (0.022)	0.127*** (0.030)	0.404*** (0.027)
Observations	645	602	575	534
Countries	42	42	40	40
Years	2000-2016	2000-2016	2000-2016	2000-2016

3) lines 35-36: reference to the "efficiency" of malaria control interventions is not clear. should one read instead "effective" when thinking about the impact that malaria interventions had on malaria control in the past?

Effective is indeed more appropriate. Thanks for this comment; this has been adjusted in the new version of the manuscript.

4) Line 42: flat funding is not the only factor that may contribute to stagnation in malaria control

Flat funding is not the only factor that may contribute to stagnation in malaria control. Our main point is to focus on aid but we did include this caveat in the discussion now.

5) line 81: not only aid increases LLIN adoption - it is likely to contribute to high level of coverage although use of interventions is less well understood.

Thanks for this clarification. In the economic literature, adoption often makes reference to ownership and use, which may have led to this confusion. We did precise this aspect in the new version.

6) eq 1: what about positive externalities of using LLIN for the "unprotected" population (those not sleeping under a net who potentially benefit from LLIN use in their community)?

It is true that the unprotected population could benefit for positive externalities but in this case, the results will not be affected as shown elsewhere (references 7, 13, 14 in the paper). We did

include one remark on this particular aspect in the discussion and added reference 13, as several important papers have shown - not only for malaria - that this could indeed be the case.

7) line 83: what is the rationale that reducing the "cost of adoption of innovations to households through aid interventions may contribute to reduce transmission"? there are certainly demand side barriers, although in the case of LLIN that have largely been distributed free of charge through mass campaign, this argument seems not well articulated with the rest of the paper. Similar comment in relation to line 123 about LLIN subsidy.

In fact, while mentioning adoption we were referring to possession and use of bednet, which might have generated the confusion here. It is not because bednets are distributed free of charge that they will be used. There might be an opportunity cost to use bednets. Moreover, aid helps to increase LLINs' coverage, which in turn may contribute to reduce transmission. However, it is true that there are certainly other demand side barriers for bednet usage (including information access, time consistency, noise around the effectiveness of LLINs).

8) equation 4: GDP per capita and international funding are closely linked as the main international funding source uses per capita income for funding eligibility and volume of aid provided - could the rationale for this specification be explained in more details?

It is true that GDP per capita and international funding are probably endogenous. Malaria and GDP per capita may also be endogenous. The purpose of the Arellano-Bond estimates is to tackle this endogeneity of malaria with respect to other issues.

As for the relationship between GDP per capita and aid per capita, it is complex and it not the focus of this paper because the table we provide does not seek to estimate the effect of GDP on malaria aid. There is a huge literature on the determinants of aid allocation and the so-called "aid dilemma" (e.g. Berthelemy, Review of Development Economics 2006) showing that aid is only partially given in response to needs. In addition, there is also a huge literature discussing the positive impact of aid on GDP. Modelling properly the interaction of aid and GDP would require another article, not particularly linked with malaria.

9) line 106: it is acknowledged by the malaria community that level of current international funding will not be sufficient to eliminate malaria - not clear what new findings are provided here.

We agree that it is acknowledged by the malaria community that level of current international funding will not be sufficient to eliminate a disease like malaria. However, firstly the models provided by the malaria community do not illustrate the possibility of a disease traps. Secondly, the advocacy on this issue has not been strongly supported by the community of economists since Sachs and Malaney (Nature, 2002) or Arrow (2004). Our paper provides renewed – theoretical and empirical - analyses on this topic, which could contribute to increase malaria aid. It also uses foundations that are more anchored in microeconomics while being related to the poverty trap literature.

10) line 131: agree that coming decades will require greater domestic ownership of health systems. Could this aspect be demonstrated by analyzing aspects of government funding and health systems required characteristics for improved malaria control?

Domestic general government health expenditure per capita have now been included in our framework and this does not affect the conclusions of our analysis (see point 2). We believe that this is the main proxy for health systems characteristics. All other aspects that are time invariant and related to health systems characteristics are taken into account through country fixed effects in the regressions but we agree that a deeper analysis of public health budget efficiency would be helpful, though this is beyond the scope of this article.

However, we agree with the referee that effectiveness is crucial. The fact that aid has a greater impact than the local public health expenditure is an indication of the weakness of the national systems - the expenditure financed by aid being certainly better controlled and administered; In order to be able to compare properly the parameters of aid and health expenditure, we also included malaria specific domestic expenditure as a share of general domestic expenditure in the model. We did not find any effect, nor the conclusions were affected. As a result, we decided to keep only general domestic health expenditure that englobe malaria specific measures for Table 1.

11) line 132. it seems important to make the case of greater coordination of current aid players such that available resources can have higher health impact. New international funding sources or innovative financing are unlikely in near future.

Thanks for this comment that is in fact crucial for the future of malaria control programs. We did insist more on this point in the new version of the manuscript and adjusted our conclusions accordingly.

Reviewer #2 (Remarks to the Author):

Thank you for your comments, which we found very useful in the revision process. Below please find our responses (your comments are in blue and italics).

If a deadly disease like Plasmodium falciparum malaria is both preventable and treatable, then it is reasonable to ask the question “why do we still have malaria”? A visual comparison of a world map of malaria burden with one of poverty will illustrate a potential answer to this question. This article succinctly examines the existence of a “disease trap”, essentially a public health version of a poverty trap for malaria. At a time when malaria financing is uncertain and recent increases in its burden threaten to derail a global campaign towards eradication, this article is both timely and important.

I think the authors are right to put a figure on the per capita level of aid required. These figures are subject to massive uncertainty, but a ballpark figure is useful for global financing discussions can be refined using microeconomic approaches at national and regional levels.

I strongly recommend the publication of this article. A few very minor points of clarification follow:

1. Preface: “whether possible to control” should be “whether it is possible to control”

Many thanks. Corrected.

2. Equations 2 & 3. The stylized model is transparent and easy to follow. I am more familiar with seeing differential equations as $dX/dt = [a \text{ function of } X \text{ and parameters}]$. Is there a reason why equations (2) and (3) are not expressed in this form?

Thanks for this comment. The equation has been changed to fit with a more standard presentation.

3. Line 75. I do not understand how $R_{0Natural}$ could be a function of H except in the mathematical sense of making it the subject of the formula for $PfRc$ given in equation (1). The basic reproduction number of any disease is defined as the expected number of new infections derived from a single infection in a fully susceptible population. For malaria it is therefore a measure of the receptivity of a setting and would not vary with changes in H . Only $PfRc$ would vary with changes in H whereas $R_{0Natural}$ would be fixed. The authors should explain this statement more clearly or remove it.

This was the meaning of our statement that $R_{0Natural}$ could be a function of H only in the mathematical sense of equation (1). We agree that this might be confusing from an epidemiological point of view and we thus removed this statement.

*4. Line 79. Solving Equation 3 at time independent equilibrium yields the solution of either $PfRc = R_{0Natural}$ (which is unstable for $\alpha > 0$) and $PfRc = R_{0Natural} * (1 - (\alpha/\beta))$, which is stable for $\alpha > 0$ and less than unity (the condition for elimination) when $R_{0Natural} * (1 - (\alpha/\beta)) < 1$. Thus, the poverty trap is for $R_{0Natural} * (1 - (\alpha/\beta)) \geq 1$. So, unless I have missed something, $PfRc$ will converge to $R_{0Natural} * (1 - (\alpha/\beta))$ not α/β as stated on line 79.*

This is true. Many thanks for your comment and we added this paragraph in the text that is perfectly clear and reflecting our view. It was indeed a typo.

5. Line 85. “reducing health aid would be catastrophic”. This statement is supported by dynamic modelling papers where a “reverse” or counterfactual scenario is explored (EG: Griffin et al. Potential for reduction of burden and local elimination of malaria by reducing Plasmodium falciparum malaria transmission: a mathematical modelling study. The Lancet Infectious Diseases, 2016). The authors may wish to refer to some of these in support of this statement in the discussion section.

Many thanks for this reference. We now refer to this particular Lancet Infectious Diseases article to support our statement.

6. Equation 4. This is a discretization of the continuous system given by equation (3). It would help to state this and to relate θ_i , δ and ϵ to the parameters of equation (3), α , etc.

Many thanks for this comment. We follow the reviewer’s recommendations on this point and we precise that Equation 4 is just a discretization of the continuous system.

7. Line 106-7. “Dynamic predictions suggest that a level of aid of US\$ 25 to \$30 per capita would be needed to reach a PfRc of 1.” What dynamic predictions? Does this come from the analysis presented or from other analyses in the literature? Please clarify.

This was coming from our own analyses, not presented in the manuscript for the sake of clarity. We prefer to keep the model as simple as possible. The paper intended to be short. Instead we preferred to quote the studies mentioned – for good reason – by the referee.

8. Line 106-7. “Dynamic predictions suggest that a level of aid of US\$ 25 to \$30 per capita would be needed to reach a PfRc of 1.” This seems about right for highly endemic African settings (Walker et al. Estimating the most efficient allocation of interventions to achieve reductions in Plasmodium falciparum malaria burden and transmission in Africa: a modelling study. The Lancet Global Health, 2016) where figures of up to US\$ 20 were estimated where elimination would be possible with current interventions, leaving some settings, predicted as requiring innovation to achieve elimination, without a predicted cost. It has been estimated that in the Asia-Pacific, where malaria incidence is lower, indicate levels of around US\$ 6 to US\$ 9 would be required for elimination (<http://www.shrinkingthemalariamap.org/sites/www.shrinkingthemalariamap.org/files/content/page/an-investment-case-for-eliminating-malaria-in-asia-pacific.pdf>).

Many thanks for the reference that we have now quoted in the new version of the manuscript in support of our results.

9. Table 1: A potential next step is to visualize the per capita investment requirements derived from the model and the MAP estimates of PfRc on a world map. Have the authors considered this? Beyond the scope of this article perhaps.

We did not consider this so far and the paper was focused on Africa only. However this is an excellent idea that we could consider for another project and will certainly try to work on this particular aspect in a near future. We will look for pluridisciplinary collaborations on this issue. Thanks for the suggestion!

REVIEWERS' COMMENTS:

Reviewer #2 (Remarks to the Author):

I feel that the points raised in my previous review have been satisfactorily addressed. I recommend that the article is published.

Lisa J White

Thank you for your comments, which we found very useful in the revision process. Below please find our responses (your comments are in blue and italics).

REVIEWERS' COMMENTS:

Reviewer #2 (Remarks to the Author):

I feel that the points raised in my previous review have been satisfactorily addressed. I recommend that the article is published.

Thank you.